# The interobserver agreement of ECG abnormalities using Minnesota codes in people with type 2 diabetes

**Giel Nijpels**[1]*, **Amber A. W. A. van der Heijden**[1], **Petra Elders**[1], **Joline W. J. Beulens**[2], **Henrica C. W. de Vet**[2]

1 Department of General Practice and Elderly Care Medicine, Amsterdam University Medical Center, Location VU, Amsterdam, The Netherlands, 2 Department of Epidemiology and Data Science, Amsterdam University Medical Center, Location VU, Amsterdam, The Netherlands

* g.nijpels@amsterdamumc.nl

## Abstract

### Objectives

To assess the interobserver agreement in categories of electrocardiogram (ECG) abnormalities using the Minnesota Code criteria.

### Methods

We used a random sample of 180 ECGs from people with type 2 diabetes. ECG abnormalities were classified and coded using the Minnesota ECG Classification. Each ECG was independently rated on several abnormalities by an experienced rater (rater 1) and by two cardiologists (raters 2 and 3) trained to apply the Minnesota codes on four Minnesota codes; 1-codes as an indication for myocardial infarction, 4 en 5-codes as an indication for ischemic abnormalities, 3-codes as an indication for left ventricle hypertrophy, 7-1-codes as an indication for ventricular conduction abnormalities, and 8-3-codes as an indication for atrial fibrillation / atrial flutter. After all pairwise tables were summed, the overall agreement, the specific positive and negative agreement were calculated with a 95% confidence interval (CI) for each abnormality. Also, Kappa's with a 95% CI were calculated.

### Results

The overall agreement (with 95% CI) were for myocardial infarction, ischemic abnormalities, left ventricle hypertrophy, conduction abnormalities and atrial fibrillation/atrial flutter respectively: 0.87 (0.84–0.91), 0.79 (0.74–0.84), 0.81 (0.76–0.85), 0.93 (0.90–0.95), 0.96 (0.93–0.97).

### Conclusion

This study shows that the overall agreement of the Minnesota code is good to excellent.

**Data Availability Statement:** All ECG results files are available from the Diabetes care System database (website: www.hoornstudies.com).

**Funding:** The authors received no specific funding for this work.

**Competing interests:** No authors have competing interests

**Abbreviations:** ECG, electrocardiogram; MI, Myocardial Infarction; IA, Ischemic abnormalities; LVH, left ventricle hypertrophy; CA, ventricular conduction abnormalities; AF, atrial fibrillation / atrial flutter; CI, confidence interval; DCS, Diabetes care System; PA, positive agreement; NA, negative agreement.

## Introduction

People with type 2 diabetes have a two-fold higher risk of cardiovascular disease than the general population [1, 2]. The resting electrocardiogram (ECG) is a simple, inexpensive, and non-invasive test that detects indications of prior myocardial infarction, ischemic abnormalities, left ventricle hypertrophy, atrial fibrillation/atrial flutter, and ventricular conduction abnormalities [3]. The recent European Society of Cardiology/European Association for the Study of Diabetes guideline recommends a resting ECG in people with type 2 diabetes and hypertension or suspected cardiovascular disease [4]. Mostly, ECGs are read centrally, with a rater not aware of clinical information. This blinded classification demands objective criteria to score for various abnormalities. The most used classification method became known as the Minnesota Code classification [5, 6]. The Minnesota Code classification consists of different code-groups, including codes as indications for myocardial infarction, ischemic abnormalities, left ventricle hypertrophy, atrial fibrillation/atrial flutter, and ventricular conduction abnormalities. The classification leads to the presence or the absence of abnormalities in different aspects of the ECG. Several studies have looked into the interrater reliability in ECG interpretation [7–12]. These studies were executed in critically ill people, people with myocardial infarction and young athletes. However, no studies have looked into the agreement measures of the Minnesota codes in people with type 2 diabetes. Because some guidelines recommend an ECG measurement in people with type 2 diabetes, it is relevant to know how well the agreement of the Minnesota codes is between different raters.

Many textbooks [13–16] recommend Cohen's Kappa as an adequate measure to reveal the level of interobserver agreement. Cohen introduced Kappa as a measure of reliability for categorical outcomes [17, 18]. The degree of agreement is more informative than the use of Kappa in clinical practice [19–21]. The terms "reliability" and "agreement" are often used interchangeably. However, the two concepts are conceptually distinct. Reliability is the ratio of variability between objects to the total variability of all measurements in the sample, the degree of agreement tells which scores or ratings are identical [22]. The agreement measures are better concepts to answer the clinical question about how colleagues would agree. The proportion of agreement distinguishes overall agreement (OA), positive agreement (PA) and negative agreement (NA). In our longitudinal cohort of people with type 2 diabetes, annual ECG measurements for over 20 years are available according to the Minnesota codes [23]. This study was interested in the interobserver agreement between different raters using the Minnesota Code classification. In other words, is the diagnosis of the different raters in agreement with each other?

## Methods

### Study design and participants

The data collection was planned after the initial Minnesota coding of rater 1. We used a computer-generated random sample of 180 participants with T2D taken during 2016 of the Hoorn Diabetes Care System study. We have described this cohort in detail elsewhere [23]. All participants gave informed consent for the anonymous use of these records for research purposes and the medical ethics committee of the VU medical center specifically approved this study and declared that an individual written consent was not needed.

### Test methods

**Resting ECG.** Resting standard 12-lead ECG was digitally acquired using a Welch Allyn electrocardiograph at 10 mm/mV calibration and speed of 25 mm/s. ECGs were initially inspected visually to exclude those with technical errors and inadequate quality before being

**Table 1. Categories of Minnesota codes.**

| Minnesota codes | code |
|---|---|
| Major QS pattern | 1–1 |
| Minor QS pattern | 1–2;1–3 |
| Tall R-wave | 3 |
| Prolonged QRS duration | 7–1;7–4 |
| ST-segment/T-wave abnormality | 4 and 5 |

automatically processed using Daxtrio software (Daxtrio medical products, Zaandam, the Netherlands).

**Minnesota classification.** ECG abnormalities were coded using the Minnesota ECG Classification, as indicated in Table 1 [6].

ECG abnormalities were coded as QS pattern (minor/major), tall R-wave, prolonged QRS duration, and ST-segment/T-wave abnormalities in the Minnesota codes system. QS patterns were considered minor if Q duration and amplitude were marginally increased (MC 1–2 and 1–3 codes), and major if Q duration and amplitude were extremely increased (MC 1–1 code), relative to the specific leads. Tall R-wave encompassed the Sokolow-Lyon-criterion or any of the following criteria: >26mm in V5 or V6; >20mm in II, III or aVF; >15 mm in I; >12mm in aVL (MC 3–1 and 3–3 codes). Prolonged QRS duration if it was a left bundle branch or intraventricular block (MC 7–1 and 7–4 codes). ST-segment/T-wave abnormalities were considered minor if ST-segments were downward sloping up to 0.5 mm below P-R baseline or if the T-wave was flat, negative, or biphasic (negative-positive type only) with less than 1.0 mm negative phase (MC 4–3 and 5–3 codes). An ST-segment/T-wave abnormality was major if an ST-segment depression with a horizontal or downward slope beyond 0.5 mm was present, or in case the T-wave was negative or biphasic (negative-positive or positive-negative type) with at least 1.0 mm negative phase (MC 4–1, 4–2, 5–1 and 5–2 codes).

**ECG rating.** Each ECG was rated by a trained physician (rater 1) and two coding cardiologists (raters 2 and 3) trained to apply the Minnesota codes. The raters had no clinical information and no knowledge of the results of the other raters. We asked the raters to score absent or present for the five Minnesota codes, 1-codes, 4- and 5-codes, 3-codes, 7-1-codes and 8-3-codes, indicating myocardial infarction, ischemic abnormalities, left ventricle hypertrophy, atrial fibrillation/atrial flutter, and ventricular conduction abnormalities. For the ischemia abnormalities, the raters also scored into major and minor abnormalities, major the 4–1, 4–2, 5–1, 5–2 codes and minor the 4–3,4–4, 5–3, 5–4 codes.

## Statistical analysis

**Method for comparing measures of agreement.** Three comparisons were possible to assess the agreement of the Minnesota codes in the case of three raters: rater 1 can be compared to rater 2 and rater 3, and raters 2 and 3 can be compared. The Overall Observed Agreement (OA), the Positive Specific Agreement (PA) and the Negative Specific Agreement (NA) can be calculated. The agreement question can be generalised as follows: given that one rater scores positive, what is the probability of a positive score by the other two raters; and the same question can also be asked for a negative score. Table 2 shows how to calculate OA, PA and NA.

The OA was calculated as the concordant cells divided by the total. The PA was calculated as two times the positive concordant cell divided by two times the positive concordant cell plus the sum of the not concordant cells. The NA was calculated as two times the negative concordant cells divided by two times the negative concordant cells plus the sum of the not concordant cells.

**Table 2. Calculation observed agreement for three raters.**

| Rater 0 vs 1 | present | absent | |
|---|---|---|---|
| present | a | b | a+b |
| absent | c | d | c+d |
| | | | a+b+c+d |
| Rater 0 vs 2 | present | absent | |
| present | e | f | e+f |
| absent | g | h | g+h |
| | | | e+f+g+h |
| Rater 1 vs 2 | present | absent | |
| present | i | j | i+j |
| absent | k | l | k+l |
| | | | I+j+k+l |

Overall Observed Agreement:

(a+e+i+d+h+l)/a to l

Specific Positive Agreement:

(a+e+i)*2/(a+e+i)*2+(b+c+f+g+j+k)

Specific Negative Agreement:

(d+h+l)*2/(d+h+l)*2 +(b+c+f+g+j+k)

To calculate the OA, PA and NA, we used the agreement formula and calculations [R package from https://github.com/iriseekhout/Agree], providing a 95% CI. To enable comparison with previous studies, we calculated the Kappa's with 95% CI using SPSS [IBM Statistics SPSS 24]. No rigid criteria were described for the OA, PA and NA. We decided to consider an excellent agreement a score of 0.81–1.00, good agreement a score of 0.61–0.80, 0.21–0.60 moderate and poor agreement a score of less than 0.20 in concordance with Kappa [20].

**Intended sample size.** Sample size calculations for reliability measures indicate that 50–100 persons are recommended when two raters are used, and aiming for the precision of CIs (confidence intervals) of +/- 0.1 and 0.2, respectively. Agreement measures correspond to these. The sample size of 180 that we used in our analysis was larger than these recommended minimum numbers [24].

## Results

All ECGs were from people with T2D, with a mean age of 68 years, and 62% were men. The mean body mass index was 29.7, the mean systolic and diastolic blood pressure respectively 146,5 and 79 mmHg, the mean HbA1c 51.6 mmol/mol, and the mean total cholesterol 4.4 mmol/l. The prevalences in this sample of myocardial infarction, Ischemic abnormalities, left ventricle hypertrophy, atrial fibrillation/atrial flutter and conduction abnormalities was 21.1% (n = 38), 48.9% (n = 88), 17.2% (n = 31), 21.1% (n = 38), and 17.8% (n = 32) respectively.

Table 3 shows the agreement proportions for the four codes and the Kappa's. Except for a low positive agreement for minor ischemic abnormalities and left ventricle hypertrophy, all agreement proportions were good or excellent.

## Discussion and conclusion

We aimed to study the concept of the agreement by more than two raters in categories of the Minnesota Code classification compared to Kappa. The results for the overall agreement were good to excellent, with values between 0.79 and 0.96. The results for the positive specific

**Table 3. The positive agreement and negative agreement scores, and Cohen Kappa's for the four ECG abnormalities based on Minnesota coding.**

| ECG abnormality | Overall agreement (95% CI) | Positive Specific agreement (95% CI) | Negative Specific agreement (95% CI) | Cohen Kappa (95% CI) |
|---|---|---|---|---|
| MI | 0.87 (0.84–0.91) | 0.69 (0.57–0.79) | 0.92 (0.89–0.95) | 0.61 (0.52–0.69) |
| IA (0/1) | 0.79 (0.74–0.84) | 0.78 (0.71–0.84) | 0.81 (0.74–0.86) | 0.59 (0.52–0.65) |
| IA–minor | 0.82 (0.78–0.86) | 0.38 (0.23–0.56) | 0.90 (0.86–0.93) | 0.51 (0.45–0.58) |
| IA–major | 0.86 (0.76–0.85) | 0.76 (0.60–0.78) | 0.91 (0.81–0.90) | 0.58 (0.52–0.69) |
| LVH | 0.81 (0.76–0.85) | 0.49 (0.34–0.64) | 0.88 (0.84–0.92) | 0.37 (0.28–0.45) |
| CA | 0.93 (0.90–0.95) | 0.80 (0.69–0.88) | 0.96 (0.93–0.98) | 0.76 (0.67–0.84) |
| AF | 0.96 (0.93–0.97) | 0.87 (0.77–0.93) | 0.97 (0.95–0.99) | 0.84 (0.75–0.92) |

Abbreviations: ECG = electrocardiogram; CI = confidence interval; MI = Myocardial Infarction; IA = Ischemic abnormalities; LVH = left ventricular hypertrophy; CA = Conduction abnormalities; AF = atrial fibrillation and atrial flutter.

agreement were good to excellent except for minor ischemic abnormalities and left ventricle hypertrophy.

When interested in interobserver agreement in clinical practice, the most relevant question is whether colleagues will provide the same answer. The proportion of observations in the same category is perhaps the most commonly used measurement to compare a set of categories. It has become customary to report the Kappa index. However, the ratio of the variability between objects to the total variability of all measurements, as Kappa does, is a reliability measure. Reliability measures are less informative in clinical practice [19]. The literature recognises the difficulty of clinical professionals in interpreting Kappa because it is a relative measure [25, 26]. That is, Kappa itself is not enough to know if professionals agree or disagree. For that reason, it is better to use agreement measures the degree to which scores or ratings are identical.

The Minnesota Code was first introduced in 1960 and subsequently extended to incorporate serial comparison in 1982 [5, 6]. Coding of an ECG can be done manually as well as using automated methods [27]. Both approaches, however, are subject to error. A 100% reliability cannot be expected on either the coding by one individual or by an automated technique [28].

The Minnesota code is a well-accepted way for categorising ECG abnormalities. The studies that looked into the reliability of the coding between different raters reported the Kappa. The reported Kappa's were similar to what we found in our study [8–12, 28, 29].

Some limitations should be kept in mind when interpreting the results of the present study. Standard criteria for the agreement measures are not available as it very much depends on the clinical use. Therefore we used the same criteria of the reliability Kappa. However, we realise that reliability and agreement are different measurement properties. In this sample, we found different prevalences of ECG abnormalities than in the total study cohort. The prevalence in the entire cohort was for myocardial infarction 13.1%, for ischemic abnormalities 29.1%, for left ventricle hypertrophy 3.4%, for conduction abnormalities 13.9%, and for atrial fibrillation/atrial flutter 11.0%. The two coding cardiologists (raters 2 and 3) did not use the Minnesota codes in the daily clinical practice. Although they were trained to apply the Minnesota codes for this study, it is evident that this could lead to miscoding and lower the agreement between the raters. We found a moderate PA for minor ischemic abnormalities and left ventricle hypertrophy. That is understandable because the coding of both abnormalities depends on minor differences in the microvolt level, and this level is challenging to measure visually.

## Conclusion

The need for comparability in clinical assessments of cardiac disease rates using an ECG as an objective measure for diagnosis and comparison led to the Minnesota codes. It seemed ideal

because it is acceptable, painless, simple, and inexpensive. This study shows that the overall agreement of the Minnesota code was good to excellent, with values between 0.79 and 0.96.

This research did not receive any specific grant from funding agencies in the public, commercial, or not-for-profit sectors.

## Author Contributions

**Conceptualization:** Giel Nijpels, Amber A. W. A. van der Heijden, Henrica C. W. de Vet.

**Formal analysis:** Giel Nijpels.

**Methodology:** Giel Nijpels, Amber A. W. A. van der Heijden, Joline W. J. Beulens, Henrica C. W. de Vet.

**Supervision:** Amber A. W. A. van der Heijden, Petra Elders, Joline W. J. Beulens.

**Validation:** Henrica C. W. de Vet.

**Writing – original draft:** Giel Nijpels.

**Writing – review & editing:** Amber A. W. A. van der Heijden, Joline W. J. Beulens.

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
