## [Decision Letter · Decision Letter 0]

18 Mar 2021

PONE-D-20-29974

The interobserver agreement of ECG abnormalities using Minnesota codes in people with type 2 diabetes.

PLOS ONE

Dear Dr. Nijpels,

Thank you for submitting your manuscript to PLOS ONE. After careful consideration, we feel that it has merit but does not fully meet PLOS ONE’s publication criteria as it currently stands. Therefore, we invite you to submit a revised version of the manuscript that addresses the points raised during the review process.

We look forward to receiving your revised manuscript.

Kind regards,

Antonio Palazón-Bru, PhD

Academic Editor

PLOS ONE

Journal Requirements:

3. Please include your tables as part of your main manuscript and remove the individual files. Please note that supplementary tables should be uploaded as separate "supporting information" files.

4. Thank you for including your ethics statement: 'Every participant have given oral permission to use their ECg results and the results were handled completely anonimous, and it is not possible to link the results to personal names or addresses. The medical ethical committee of the VU medical center gives permission to handle the data as we did.'

(a) Please amend your current ethics statement to confirm that your named institutional review board or ethics committee specifically approved this study.

(b) Once you have amended this/these statement(s) in the Methods section of the manuscript, please add the same text to the “Ethics Statement” field of the submission form (via “Edit Submission”).

5. In the Methods, please clarify that participants provided oral consent. Please also state in the Methods:

- Why written consent could not be obtained

- Whether the Institutional Review Board (IRB) approved use of oral consent

- How oral consent was documented

6. To comply with PLOS ONE submission guidelines, in your Methods section, please provide additional information regarding your statistical analyses. For more information on PLOS ONE's expectations for statistical reporting, please see https://journals.plos.org/plosone/s/submission-guidelines.#loc-statistical-reporting.

7. We note that you have stated that you will provide repository information for your data at acceptance. Should your manuscript be accepted for publication, we will hold it until you provide the relevant accession numbers or DOIs necessary to access your data. If you wish to make changes to your Data Availability statement, please describe these changes in your cover letter and we will update your Data Availability statement to reflect the information you provide.

Reviewers' comments:

Reviewer's Responses to Questions

**Comments to the Author**

1. Is the manuscript technically sound, and do the data support the conclusions?

Reviewer #1: Partly

2. Has the statistical analysis been performed appropriately and rigorously? 

Reviewer #1: Yes

3. Have the authors made all data underlying the findings in their manuscript fully available?

Reviewer #1: Yes

4. Is the manuscript presented in an intelligible fashion and written in standard English?

Reviewer #1: No

5. Review Comments to the Author

Reviewer #1: Thank you very much for the invitation to review this manuscript. Estimating measurement errors is important because it affects the internal validity of study results.

Introduction and general: Using abbreviations is fine, but because there are so many in this manuscript, it makes it sometimes very hard to read. I would strongly recommend to reduce the number of abbreviations substantially and to keep the most relevant.

Introduction, page 3, first paragraph: I recommend to use more precise language. For example, a classification cannot lead to something, it may classify things into categories. What do you mean by ‘epidemiological purpose’?

Introduction and general: Please be more specific regarding the measurement properties you are referring to. In the introduction the concepts ‘agreement’, ‘reproducibility of agreement’, ‘reliability’ seem to be used interchangeably. Then the text says ‘how well …. codes can be classified…’ which is closely connected to diagnostic accuracy. I would strongly recommend to adopt commonly accepted terminology such as the COSMIN framework. There is a fundamental difference between reliability and agreement, and both measure different aspects. Please note that the coefficient kappa is a reliability coefficient and it is very informative, because it says something about the distinguishability of scores or categories (e.g. doi: 10.1016/j.jclinepi.2010.03.002 or doi: 10.1016/j.jclinepi.2010.12.001). (Specific) agreement measures something else (see COSMIN).

Methods and results: I recommend to structure these sections according a suitable reporting guideline. Please provide much more demographic data about the patients and raters. Please provide much more details about the ECG procedure, who did this, where, when etc. How was a random sample selected. How was the sample size determined?

Methods, Minnesota classification: It is impossible to understand the second sentence. Please explain step by step how this instrument looks like, please consider a table or figure and explain to readers the coding.

Methods, statistical analysis: Please explain what you mean by ‘summed’. Please use past tense consistently. The agreement estimates you obtained maybe considerd as proportions or ratios, but not ‘scores’. Because reliability (kappa) and agreement estimates are totally different measurement properties, you cannot use the same evaluation about the measurement properties. Again, the COSMIN framework provides guidance about how to evaluate measurement properties and I strongly recommend to follow an accepted approach (e.g. DOI: 10.1186/s13063-016-1555-2).

Results: Was it the same three raters who rated all 180 ECGs? The results section is largely a repetition of the table and difficult to read.

Discussion, page 6, first paragraph: Please revise the sentence ‘The proportions classified correctly …’ In reliability and agreement you never know what was ‘correct’. You measure the relative and absolute measurement properties, not the diagnostic accuracy. Please carefully reconsider the use of concepts for measurement properties (see above). Please revise ‘rate of agreement’.

Conclusion: Please base your conclusions only on the results you provide. Currently it goes much beyond of what you did. The results do indicate that specific agreement was sometimes very low. Why do you conclude that this is acceptable? Please define what you consider as ‘acceptable’ before in the methods. In fact, specific agreement heavily depends on the prevalence and therefore it behaves often very similar to kappa. This can be nicely seen in Table 2 when comparing the third and fifths column. Therefore, I recommend to base your interpretation mainly on overall agreement (because it measures the absolute measurement error without any assumptions) and kappa as a ‘simple’ reliability coefficient. You may use proportions of specific agreement to look into particular classification details and rater behaviour. If you follow this recommendation, please adjust the manuscript accordingly.

Abstract: Please delete any abbreviations and revise after the main text has been revised.

6. PLOS authors have the option to publish the peer review history of their article (what does this mean?). If published, this will include your full peer review and any attached files.

Reviewer #1: **Yes: **Jan Kottner

---

## [Author Response · Author response to Decision Letter 0]

4 May 2021

Reviewer comments:

Introduction and general: Using abbreviations is fine, but because there are so many in this manuscript, it makes it sometimes very hard to read. I would strongly recommend to reduce the number of abbreviations substantially and to keep the most relevant.

We have reduced most of the abbreviations and have kept only the most relevant. 

Introduction, page 3, first paragraph: I recommend to use more precise language. For example, a classification cannot lead to something, it may classify things into categories. What do you mean by 'epidemiological purpose'?

We recognise that the language was not very precise and changed that accordingly. We have removed the sentence about epidemiological purpose because that was redundant.

Introduction and general: Please be more specific regarding the measurement properties you are referring to. In the introduction the concepts' agreement', 'reproducibility of agreement', 'reliability' seem to be used interchangeably. Then the text says 'how well …. codes can be classified…' which is closely connected to diagnostic accuracy. I would strongly recommend to adopt commonly accepted terminology such as the COSMIN framework. There is a fundamental difference between reliability and agreement, and both measure different aspects. Please note that the coefficient kappa is a reliability coefficient and it is very informative, because it says something about the distinguishability of scores or categories (e.g. doi: 10.1016/j.jclinepi.2010.03.002 or doi: 10.1016/j.jclinepi.2010.12.001). (Specific) agreement measures something else (see COSMIN).

We have changed the wording according to the COSMIN guidelines throughout the whole text. 

Methods and results: I recommend to structure these sections according a suitable reporting guideline. Please provide much more demographic data about the patients and raters. 

We have followed the guidelines for Reporting Reliability and Agreement Studies (GRRAS guidelines) for the different sections of the manuscript. We have provided more details about the participants in the study (page 4, line 219). 

All ECGs were from people with T2D, with a mean age of 68 years, and 62% were men. The mean body mass index was 29.7, the mean systolic and diastolic blood pressure respectively 146.5 and 79.0 mmHg, the mean HbA1c 51.6 mmol/mol, and the mean total cholesterol 4.4 mmol/l. The prevalences in this sample of myocardial infarction, Ischemic abnormalities, left ventricle hypertrophy, atrial fibrillation/atrial flutter and conduction abnormalities were 21.1% (n=38), 48.9 % (n=88), 17.2 % (n=31), 21.1 % (n=38), and 17.8 % (n=32), respectively. 

Table 2 shows the agreement proportions for the four ECG codes and the Cohen Kappa's. Except for a low positive agreement for minor ischemic abnormalities and left ventricle hypertrophy, all agreement proportions were good or excellent.

Please provide much more details about the ECG procedure, who did this, where, when etc. How was a random sample selected. How was the sample size determined?

On page 4, line 149, we give more details about the selection of the participants.

The data collection was planned after the initial Minnesota coding of rater 1. We used a computer-generated random sample of 180 participants with T2D taken during 2016 of the Hoorn Diabetes Care System study.The sample size calculation is described on page 5 line 210. 

Intended sample size: Sample size calculations for reliability measures indicate that 50-100 persons are recommended when two raters are used, and aiming for precision of confidence intervals of +/- 0.1 and 0.2, respectively. Agreement measures correspond to these. The sample size of 180 that we used in our analysis are larger than these recommended minimum numbers (de Vet HCW, Terwee CB, Mokkink LB. Knol DL. Measurement in Medicine, Practical guides to Biostatistics and Epidemiology. Cambridge University Press 2011, Cambridge).

Methods, Minnesota classification: It is impossible to understand the second sentence. Please explain step by step how this instrument looks like, please consider a table or figure and explain to readers the coding.

We have explained step by step how the Minnesota codes are defined and scored on page 4 line 163-176. We also added a table (table 1).

ECG abnormalities were categorised in the Minnesota codes system as QS pattern (minor/major), tall R-wave, prolonged QRS duration, and ST-segment/T-wave abnormalities. QS patterns were considered minor if Q duration and amplitude were marginally increased (MC 1-2 and 1-3 codes), and major if Q duration and amplitude were extremely increased (MC 1-1 code), relative to the specific leads. Tall R-wave encompassed the Sokolow-Lyon-criterion or any of the following criteria: >26mm in V5 or V6; >20mm in II, III or aVF; >15 mm in I; >12mm in aVL (MC 3-1 and 3-3 codes). Prolonged QRS duration if it was a left bundle branch or intraventricular block (MC 7-1 and 7-4 codes). ST-segment/T-wave abnormalities were considered minor if ST-segments were downward sloping up to 0.5 mm below P-R baseline or if the T-wave was flat, negative, or biphasic (negative-positive type only) with less than 1.0 mm negative phase (MC 4-3 and 5-3 codes). An ST-segment/T-wave abnormality was major if an ST-segment depression with a horizontal or downward slope beyond 0.5 mm was present, or in case the T-wave was negative or biphasic (negative-positive or positive-negative type) with at least 1.0 mm negative phase (MC 4-1, 4-2, 5-1 and 5-2 codes).

Methods, statistical analysis: Please explain what you mean by 'summed'. Please use past tense consistently. The agreement estimates you obtained maybe considered as proportions or ratios, but not 'scores'. Because reliability (kappa) and agreement estimates are totally different measurement properties, you cannot use the same evaluation about the measurement properties. Again, the COSMIN framework provides guidance about how to evaluate measurement properties and I strongly recommend to follow an accepted approach (e.g. DOI: 10.1186/s13063-016-1555-2).

We thank the reviewer for this valuable remark. We have changed a part of the introduction on page 4, line 64-70. 

The terms "reliability" and "agreement" are often used interchangeably. However, the two concepts are conceptually distinct. Reliability is the ratio of variability between objects to the total variability of all measurements in the sample. The degree of agreement tells which scores or ratings are identical. We have removed summed, and the calculation is explained on page 5, line 211-217. We changed the present tense to the past tense, where eligible. We did not find any evaluation criterion for the agreement measures also not in the paper of Prinsen et al. (How to select outcome measurement instruments for outcomes included in a "Core Outcome Set" - a practical guideline). These are also not mentioned in the GRRAS paper. Another paper by Prinsen (on COSMIN) mentions criteria for measuring continuous outcomes but not agreement percentages for categorical outcomes. Therefore, we decided to use the reliability (Kappa) estimates criteria, although we realise that agreement and reliability different measurement properties. In the discussion, we add a sentence under the limitation paragraph. Page 7 line 304-307.

Standard criteria for the agreement measures are not available as it very much depends on the clinical use. Therefore we used the same criteria of the reliability Kappa; however, we realise that reliability and agreement are different measurement properties. 

Results: Was it the same three raters who rated all 180 ECGs? The results section is largely a repetition of the table and difficult to read.

In the result section, we have clarified the raters of the ECG. We removed the numbers from the table and simplified the text on page 6, line 247-255.

All ECGs were from people with T2D, with a mean age of 68 years, and 62% were men. The mean body mass index was 29.7, the mean systolic and diastolic blood pressure respectively 146.5 and 79.0 mmHg, the mean HbA1c 51.6 mmol/mol, and the mean total cholesterol 4.4 mmol/l. The prevalence in this sample of myocardial infarction, Ischemic abnormalities, left ventricle hypertrophy, atrial fibrillation/atrial flutter and conduction abnormalities was 21.1% (n=38), 48.9 % (n=88), 17.2 % (n=31), 21.1 % (n=38), and 17.8 % (n=32), respectively. 

Table 2 shows the agreement proportions for the four codes and the Kappa's. Except for a low positive agreement for minor ischemic abnormalities and left ventricle hypertrophy, all agreement proportions were good or excellent.

Discussion, page 6, first paragraph: Please revise the sentence 'The proportions classified correctly …' In reliability and agreement you never know what was 'correct'. You measure the relative and absolute measurement properties, not the diagnostic accuracy. Please carefully reconsider the use of concepts for measurement properties (see above). Please revise 'rate of agreement'.

We changed the sentence and removed classified correctly in: The proportion of observations in the same category is perhaps the most commonly used measurement to compare a set of categories. In the whole manuscript, we carefully revised the wording of the measurement properties. 

Conclusion: Please base your conclusions only on the results you provide. Currently it goes much beyond of what you did. The results do indicate that specific agreement was sometimes very low. Why do you conclude that this is acceptable? Please define what you consider as 'acceptable' before in the methods. In fact, specific agreement heavily depends on the prevalence and therefore it behaves often very similar to kappa. This can be nicely seen in Table 2 when comparing the third and fifths column. Therefore, I recommend to base your interpretation mainly on overall agreement (because it measures the absolute measurement error without any assumptions) and kappa as a 'simple' reliability coefficient. You may use proportions of specific agreement to look into particular classification details and rater behaviour. If you follow this recommendation, please adjust the manuscript accordingly.

We assume that the reviewer means the discussion, including the conclusion paragraph. As already stated before, we did not find any evaluation criterion for the agreement measures. Therefore we decided to use the criteria of the reliability (kappa) estimates, although we realise that agreement has different measurement properties. For that reason, we concluded that the observed agreement proportions were as described. We changed the last sentence in the conclusion section.

This study shows that the overall agreement of the Minnesota code was good to excellent, with values between 0.79 and 0.96. Abstract: Please delete any abbreviations and revise after the main text has been revised.

We have deleted all the abbreviations in the abstract.

---

## [Editor Report · Decision Letter 1]

19 Jul 2021

The interobserver agreement of ECG abnormalities using Minnesota codes in people with type 2 diabetes.

PONE-D-20-29974R1

Dear Dr. Nijpels,

We’re pleased to inform you that your manuscript has been judged scientifically suitable for publication and will be formally accepted for publication once it meets all outstanding technical requirements.

Kind regards,

Antonio Palazón-Bru, PhD

Academic Editor

PLOS ONE
---

## [Editor Report · Acceptance letter]

26 Jul 2021

PONE-D-20-29974R1 

The interobserver agreement of ECG abnormalities using Minnesota codes in people with type 2 diabetes. 

Dear Dr. Nijpels:

I'm pleased to inform you that your manuscript has been deemed suitable for publication in PLOS ONE. Congratulations! Your manuscript is now with our production department. 

Kind regards, 

on behalf of

Dr. Antonio Palazón-Bru 

Academic Editor

PLOS ONE